# Challenges in making standardisation work in healthcare: lessons from a qualitative interview study of a line-labelling policy in a UK region

Natasha Marie Kriznik , Guillaume Lamé , Mary Dixon-Woods

THIS Institute (The Healthcare Improvement Studies Institute), Department of Public Health and Primary Care, University of Cambridge, Cambridge, UK

**Correspondence to**
Professor Mary Dixon-Woods; director@thisinstitute.cam.ac.uk

## ABSTRACT

**Objective** To identify and learn from efforts to design and implement a standardised policy for labelling of invasive tubing and lines across a regional health system.

**Design** Single case study involving qualitative interviews and documentary analysis.

**Setting** A devolved health system in the UK National Health Service (NHS).

**Participants** NHS staff (n=10) and policy-makers (n=8) who were involved in the design and/or implementation of the standardised policy.

**Results** Though standardising labelling of invasive tubing and lines was initially seen as a common-sense technical change, challenges during the process of developing and implementing the policy were multiple and sociotechnical in nature. Major challenges related to defining the problem and the solution, limited sustained engagement with stakeholders and users, prototyping/piloting of the solution, and planning for implementation. Some frontline staff remained unconvinced of the need for or value of the policy, since they either did not believe that there was a problem or did not agree that standardised labelling was the right solution. Mundane practical issues such authorisation and resourcing, supply chains for labels, the need to restructure work practices to accommodate the new standard, and the physical features of the labels in specific clinical settings all had important impacts.

**Conclusions** Newly standardised tools and practices have to fit within a system of pre-existing norms, practices and procedures. We identified a number of practical, social and cultural challenges when designing and implementing a standardised policy in a regional healthcare system. Taking account of both sociocultural and technical aspects of standardisation, combined with systems thinking, could lead to more effective implementation and increase acceptability and usability of new standards.

### Strengths and limitations of this study

► The case study design allowed for an in-depth exploration of the development and implementation of a regional policy for standardising labelling of patient lines and tubes.
► We were able to recruit a mixture of policy-makers and frontline staff which allowed us to understand a range of different experiences of the design and implementation of the regional line-labelling policy.
► We were not able to interview as many people as originally planned, and the geographical spread of those interviewed was uneven.
► The study used interviews and documentary analysis; direct observations were not possible.

## INTRODUCTION

Standardisation is often considered as an intuitively obvious solution to some types of challenges facing the quality and safety of healthcare.[1] Broadly defined, standards are specifications of rules, guidelines or characteristics for designing products or carrying out activities.[2] Standardisation has a promising role in reducing error,[3–5] removing unnecessary variation[4 6] and reducing uncertainty in clinical interactions.[7] Despite the increasing penetration of guidelines and other forms of standardised practices into healthcare, many everyday objects, patient pathways, workflows and tools of work remain highly localised and variable from setting to setting.[8] Sometimes this variability is appropriate; sometimes, however, it introduces risks and inefficiencies that would be best addressed through standardisation or harmonisation. One example was the entropy of crash call numbers across the National Health Service (NHS) which persisted until the introduction of the standard 2222 number in 2004.[9] However, standardisation is rarely a straightforward process: challenges may arise, for example, because standards themselves are defined and implemented through complex sociotechnical processes involving people, routines, protocols and technologies, requiring multiple kinds of purposeful effort during development, implementation and maintenance stages.[10] Understanding the existing system into which standards are to be introduced or refined is crucial for their success (ie, for their uptake

by stakeholders).[11] Yet the challenges associated with standardisation of everyday objects have remained largely neglected as a focus of study.

In this article, we report a qualitative case study of the development and implementation of a line-labelling policy that sought to standardise labelling of invasive tubing and lines across a single devolved regional health system in the UK. Invasive tubes and lines (which we will refer to collectively as 'lines') are used for multiple purposes in patient care, including patient monitoring, administration of fluids, medication or feeding, and drainage of bodily fluids. Though ubiquitous in healthcare, lines are associated with significant safety challenges, including those relating to wrong route and wrong site error and infection[12 13] and the risks may be increased when a large number of lines are positioned in close proximity to one another or when patients move from one service to another.[14] Consistent labelling of lines is considered good practice and, in principle, has potential to mitigate some of these problems.[15–17] Some countries, such as Australia, have issued a national policy to standardise labelling of lines and injectable medicines,[18] but, notwithstanding some guidance from the now-defunct National Patient Safety Agency,[19] the UK has not done so.

It is important that attempts to standardise cumulate learning. We aimed to examine a recent policy effort to design and implement a standardised line-labelling policy in one of the four devolved nations of the UK, with the aim of learning from the experience and informing practical ways to improve similar efforts in the future.

## METHODS

We conducted a single case study[20] of the development and implementation of a line-labelling policy in one of the four nations of the UK with a devolved health system. The case study design allowed for an in-depth exploration of the project and its context, based on interviews with a range of clinical and non-clinical staff in local trusts (NHS organisations) and documentary analysis. The organisations and individuals participating in the case study are anonymised.

Primary data came from semi-structured telephone interviews with NHS staff and policy makers, conducted between February and September 2017, following the introduction of the line-labelling policy in January 2017. Key staff members were identified by the project lead for the regional line-labelling policy, and those staff were contacted through the chief medical officer's (CMO's) office with an invitation to take part in an interview, an information sheet explaining the nature of the study and information to allow them to respond direct to the study team. In total, 64 staff members were invited. The study team received 27 responses from interested participants. Due to scheduling difficulties and some lack of response to follow-up contact, we completed 18 interviews. Interviews were audio recorded after obtaining written consent and transcribed verbatim.

In addition to the interviews, we analysed documentary sources related to the line-labelling policy which were provided to the study team by the project lead and dated from 1998 onwards. These sources comprised minutes of meetings that discussed reviews of line labelling policies from one trust; minutes of meetings from the Regional Policy Group discussing development of plans for region-wide standardisation; presentations to staff about practices of line labelling; a copy of the letter from the CMO to staff about introducing the new policy; previous local line-labelling policies dating from 1998 and 2008 from one trust; and the text of the new regional policy, including a poster presenting the label designs. These documents helped the study team to understand the history of the policy's development which is outlined in the first section of the Results.

Both interview transcripts and documentary data analysis was based on the constant comparative method,[21] and NVivo (QSR V.11) was used in order to organise and manage data and coding effectively. The analysis was concurrent with data collection and was carried out by NMK. Coding was largely inductive, but was also informed by sensitising concepts from the quality improvement and organisational change literature. Codes were then organised into four global themes, reflecting key dimensions of the process of developing and implementing a new standard: operational and organisational considerations, adaptive considerations, technical considerations and implementation considerations.

### Patient and public involvement

This was a study of a policy targeting staff behaviours and organisational systems. Patients and the public were not involved in this study.

### Data availability

The consent for this project restricts use of the transcripts to the study team. Requests to access anonymised data should be made to the corresponding author; further ethical and governance approvals may be required to enable sharing.

## RESULTS

Of the 18 individuals interviewed, 10 were policy-makers (eg, civil servants) who were part of a working group charged with designing and overseeing the implementation of the policy. Eight were NHS frontline staff who were involved in implementing the policy across three trusts (healthcare organisations).

We offer, first, a descriptive summary of the historical development of the devolved region's standardised line-labelling policy using primarily documentary sources, before highlighting key challenges during the process of designing and implementing region-wide standardisation.

### Developing and implementing the policy

Some of the origins of the policy lay partly in one of the trusts in the region, which had, in 1998, developed

its own local line-labelling policy. It provided guidance aimed at ensuring that invasive lines could be traced to their source and were labelled appropriately, using nine types of standardised labels. After a merger, the policy was updated in 2008 and extended to 11 types of labels. An audit in the trust in April 2011 demonstrated poor compliance with the 2008 policy. Staff reported that they could not locate the specific labels required by the policy, and that multiple different labels were available for the same line. It also became apparent that line-labelling practices—for example, whether lines were labelled at all, and how they were labelled—varied across the trust. These variations were seen by trust management as concerning.

> An audit took place in Paediatrics; the objective being to ensure the policy is being adhered to and to improve compliance if appropriate, the overall result was very poor as they were unable to obtain the coloured labels stated in the policy. (Minutes from Standards and Guidelines Committee Meeting in one trust, April 2011)

The trust brought these concerns to the Regional Policy Group Collaborative (RPGC), a committee established in 2010 to coordinate the production of clinical policy across the health system. At a meeting of the RPGC in December 2011, it was identified that these concerns were likely to be evidence of a more generalised problem, and it was agreed that a region-wide line-labelling policy should be developed and implemented.

> [Two trusts] reported that there are few national standards [for line labelling]. Agreement from group that we should make an effort to deliver a regional policy. (Minutes from RPGC meeting, December 2011)

In March 2012, a working group comprising representatives from both the RPGC and each trust in the health system was established to oversee the development of the new policy, with two project leads. The working group completed a scoping exercise to understand what line-labelling policies and label designs were already in use in different trusts. It then produced a first draft of the line-labelling policy and submitted it to the RPGC for comment. In December 2012, the working group received some initial feedback about amending the title of the policy and the need for all labels to be coloured. At the following RPGC meeting in March 2013, members agreed that the policy's scope should be widened to include all possible lines, not just invasive lines (as had been the case in previous trust-level policies), and that each label should have its own colour code.

> [Project lead] to amend and widen the scope of the policy.
>
> The Collaborative agreed the list should include as many tubes/lines as possible and assign and [a] colour [to] each.
>
> (Minutes from RPGC meeting, March 2013)

The working group consulted with frontline staff across the health system (senior doctors, nursing leads and other senior staff) via email to get their feedback on the policy, with the aim of achieving consensus on the designs for the labels (colours, fonts and additional identifying features such as borders), and on the design for a poster for use in the clinical settings where staff would be using the labels.

> Each representative on the Collaborative will email MDs/Nursing Leads and Senior Staff to agree colour before ordering. (Minutes from RPGC meeting, March 2013)

The feedback was incorporated into the final policy, which had 21 different types of labels. In addition to setting out a colour code for each line and a standard catalogue of label designs, the policy also sought to standardise practices associated with line-labelling, requiring that all lines be labelled and that all staff (with the explicit exception of ambulance staff) label lines. The primary purpose expressed in the policy document was 'to safeguard the patient and health care professional by reducing 'wrong route administration' area of risk and to avoid confusion where invasive lines and tubes are in situ' (Policy for the identification and labelling of invasive lines and tubes, January 2017).

It is unclear exactly when the final draft of the policy was agreed, in part because the RPGC was dissolved at some point during the policy's final development stages. "Months, maybe a year, or more" (participant 16, working group) passed between the moment when the final draft policy was agreed by the working group and the policy was adopted by senior policy-makers.

During 2016, the health system's CMO agreed that the policy should be adopted. In November 2016, the CMO and the chief nursing officer (CNO) sent a joint letter to all trusts informing them that the policy would be introduced in early January 2017. All trusts were expected to implement the policy at the same time. They were asked to remove old labels from their stores and to ensure that they had ordered enough new labels to arrive in time for the proposed date. However, the start date had to be pushed back until the end of January 2017, following problems obtaining a financial contribution from each trust to cover the cost of printing plates for the new labels. The CMO and CNO then issued a second joint letter notifying trusts of the new date of implementation once the issue had been resolved.

> There is no phased introduction of the policy—all parts of every organisation are expected to comply from 9 January. Please ensure that any previous drafts of the policy are removed and replaced with the version included with this letter. (Extract from joint letter from CMO and CNO, November 2016)

## Challenges encountered

Four major challenges were encountered, relating to defining the problem, limited sustained engagement with

stakeholders and users, prototyping/piloting of the solution and planning for implementation.

### Defining the problem and the solution

There was reasonable consensus among participants that the main aim of the region-wide line-labelling policy was to improve patient safety by reducing the risk of wrong route administration occurring, that is, "To help them [staff] not to do the wrong thing" (participant 4, working group).

> …the aim of the line-labelling programme, as far as I can see, was to make sure that, number one, that there was standardisation across the region, so that when nurses and doctors move between hospitals, the colours and the design of the labels is the same, so you're less likely to…have harm or error done. (Participant 11, working group)

As one member of the working group put it, standardising line-labelling was seen as "a no-brainer" (participant 7). Some members of the working group felt that the need to standardise line-labelling in order to prevent wrong route administration was so obvious that they did not need to convince frontline staff of this.

> I suspect we didn't really need to persuade people that it was a patient safety issue. It was a fairly obvious one. (Participant 2, working group)

But, from the beginning, the working group seemed to focus on the solution (standardising line-labelling across the region) rather than on the actual problem (wrong route administration). Though the working group had some anecdotal evidence on the incidence of wrong route administration, no quantified baseline data were available, nor was evidence explicitly linking the problem to inadequate line-labelling.

> One of the things that we didn't have any baseline for…was any incidents to quantify the risks of it not being standardised. I suppose it seemed to be more of a sense that this was the right thing to do rather than any evidence that people had come to harm because it wasn't done. (Participant 5, working group)

Staff at the sharp end of care did not universally share the view that there was in fact a problem or that the standardisation policy was the solution.

> I am a little unclear…as to exactly what the aim and benefit of the line-labelling programme is. …I suppose I don't really feel anybody has necessarily ever explained the need, the need for the policy…I don't think anybody has really made the case for the need for change. (Participant 12, frontline)

Staff at the sharp end raised several concerns about the effectiveness of labels in preventing wrong route administration from happening. For example, the same interviewee felt that "it's a partial system, in that, if someone is going to put the wrong substance into the wrong line, having a sticker on it, it doesn't force them in any way". A senior nurse was "not convinced that a label actually reduces harm to patients" (participant 6, frontline). A nurse from a dialysis unit recalled an incident where a doctor misused a dialysis catheter, but where a label would not have helped matters:

> …to have a label on that dialysis catheter would not have prevented that doctor from doing what he did… He knew it was a dialysis catheter, but he just decided that he couldn't get peripheral access and that he would go ahead and that he would use this for blood and that he would deliver the antibiotics down it, he didn't use (antiseptic technique) and he didn't flush it properly and it clotted. But a label wouldn't have prevented that. …he already knew it was dialysis, the patient told them, this is my dialysis line, this is my dialysis catheter. (Participant 17, frontline)

### Limited sustained engagement with stakeholders and users

The members of the working group involved in the design and development stages gave a generally positive account of the consultation with staff. They emphasised that representatives for frontline staff in different trusts and professional groups had helped to develop the policy as part of the working group, and other staff were consulted about the label designs.

> It's not like something that's been put upon them with no consultation, you know, this has never been put upon anybody, everybody had the opportunity to consult. …we got lots of comments back and we took some of them on board and others we sort of put a line through them. But I think that's why it's going to work because we touched so many stakeholders involved and so many people consulted. (Participant 1, working group)

> It's about ensuring that they're involved from the very start. So they were the key people round the table, the staff who would actually have to take it back to their organisation and say, this is what we want to do. (Participant 8, working group)

However, frontline staff perceived things somewhat differently, reporting that when they were emailed for their feedback on the policy, they were sent a copy of the poster that set out the label designs and were asked for their comments on it. They reported that they did not have a say on the problem to be addressed or on the solution set out in the policy, only on the details of its implementation. Some staff we interviewed at the sharp end were sceptical about the consultation process as a result, complaining, for example, about short turnaround times.

> I received [the poster with label designs] 1 day and a few days later, it almost had to be back within two or 3 days, so I didn't follow that up physically. I didn't walk around everywhere asking people to look at

it, and so I got no comments back. (Participant 14, frontline)

Another saw the consultation as more informational than consultative.

It didn't come as a finished product but you didn't have input either. It was very much that this was a work in progress and that there were people from every trust meeting regionally to discuss, and that they would then be implementing it. (Participant 17, frontline)

### Prototyping/piloting of the solution

The working group thought the policy was too straight-forward to justify experimentation with designs and physical examples of the labels, especially as it had assumed that all staff were already labelling all lines, although inconsistently.

This was merely a state of change and once we'd agreed the labels it was just a case of swapping them over. (Participant 5, working group)

The absence of prototyping and piloting meant that practical issues with the policy emerged only after implementation. Many of these issues related primarily to the label designs. For instance, one participant pointed out that the same colour label was used for two different lines, and that the colours chosen for some labels did not match existing colouring systems used for identifying devices or injectable medications for specific lines, suggesting that the scoping exercise and consultation with staff on the label designs had not been sufficiently comprehensive.

I just know with the epidural labels, in the epidural policy that we have for the trust, everything's supposed to be yellow but not only does it [the new epidural label] match the colour for the pulmonary artery flotation catheter, it's not actually yellow, it's not what staff would expect either, it's like…a yellowy green colour, and then the enteral tube labels are supposed to be purple, so it's not just that they match or are similar to other labels on the chart, but they're not the colour that they should be in the first place. (Participant 14, frontline)

The same participant also commented that the new labels were "not of the same quality and they just look more shabby" than labels they had used previously. These apparently lower quality labels required changing more frequently, but were also stickier, which caused problems when changing labels on paediatric patients, particularly neonates. The size of the labels also caused problems in both paediatric and adult wards.

…the size of the labels themselves, they're large, and I have had a number of emails…around the size of the labels and how that's interfering sometimes with…one of them was a patient's NG [nasogastric] tube. The size of the label meant that it was obscuring

the vision of the patient because of…the position of the label that's required in the policy. (Participant 6, frontline)

Additionally, line-labelling was a new practice in some specialties, contrary to the working group's assumption that all staff were already labelling lines. In dialysis, for example, lines were not regularly labelled, as staff reported that dialysis lines were designed for a specific purpose and could not be confused for anything else (as noted in the example above with the doctor who misused a dialysis catheter), so labelling them was unnecessary work for staff.

…we feel it's kind of one policy fits all and it really doesn't fit with us for a number of reasons. We're labelling lines that we know exactly what they are; every single person who's working in that unit knows what dialysis lines are. (…) People are mentored for 6 weeks and trained how to put these lines on – you cannot mistake these lines for anything else, they are dialysis machine lines, they won't fit anything else, there's slots on them that will only fit into a machine (…) [I]t's a nonsense for something that has no value, whether you write catheter, whether you write line, everybody knows these are dialysis lines. There's no value whatsoever, and it's just adding work for people. (Participant 17, frontline)

### Planning for implementation

Obtaining appropriate sign-off for the policy before implementation proved challenging. The dissolution of the RPGC meant that the working group struggled to identify how to get the policy approved.

…we had [the RPGC], and that would have, probably, been the place that things like this would have been signed off, and that group ceased to exist…there were delays in trying to get…what would replace that, what's the mechanism for getting regional sign off, so things, kind of, went quiet for a while. (Participant 16, working group)

In one trust, the policy was only formally signed off in December 2017 to be implemented from January 2018, when the initial objective was to implement in January 2017.

Sites followed instructions to remove old labels ahead of the regional implementation date in January 2017, but issues with printing and distributing the labels led to delays. A small amount of funding was needed to create the new printing plates, but this turned out to be difficult to source from each of the trusts, delaying the project.

We did get into a ludicrous situation at one point… we needed a small amount of money, it was less £1000 and trying to find less than £1000 to buy your first batch of labels was horrendously difficult. (Participant 5, working group)

Further, the number of labels needed for the first order was underestimated. While these problems were dealt with relatively quickly, not all sites that were due to implement the policy had received their labels on the new start date.

> Well, when they [the labels] initially came here there was a problem in stock…the first week we started we were on the back foot and told that we really didn't have the supply of labels that we probably would need. But we've eventually sorted that and it's not a problem now, but they just hadn't really anticipated the demand that we would require, so that was a bit disappointing that someone hadn't thought that one through really. (Participant 3, frontline)

Issues with the supply chain of labels from the printer to the wards lasted for at least 6 months in some sites. Although one of the working group leads stated in February 2017 that "the issue has now been resolved" (participant 1, working group), one hospital nurse, interviewed in June 2017, had still not received all the labels for their department.

The working group expected managers to be key stakeholders of the policy and support implementation. However, not all managers were behind the policy, and not all ensured that the policy was implemented in their organisation or that staff seeking to implement the policy were adequately supported.

> Well, the chief medical officer gave a presentation… and it was clear that [organisations within] the trust were not all implementing the labels, some I think hadn't…so he did emphasise the need for medical directors in the trust to take ownership of this and make sure it all gets implemented properly. (Participant 15, working group)

> I think support from senior level in the trust would be helpful because now that people have reported concerns to me I need to reassure them that not only have I passed that on, but that actions are going to be taken or whatever, so that's important for people at clinical level to get that feedback. (Participant 14, frontline)

The working group planned that an initial audit, led by a regional quality improvement authority, should take place three to 6 months after the initial start date to evaluate the implementation of the policy, focusing mainly on staff compliance. Although reducing wrong route administration was the primary purpose of the policy, the impact of the policy could not be assessed as no baseline data on this issue were collected before the policy was implemented.

## DISCUSSION

This study has identified that many of challenges faced throughout developing and implementing a standardised line-labelling policy across a single devolved health system were practical and logistical in nature, while others were more social and cultural in character. These findings have broader relevance for large-scale standardisation exercises.

Convincing people that there is a problem, and that the proposed solution is appropriate, are known challenges in healthcare improvement.[22] This project ran into challenges relating to problem definition[23]: the working group believed the problem and the solution they proposed were obvious and compelling, but this view was not shared across all levels of the system. The lack of baseline data to support the project in the first place seemed to be a key driver here; indeed, recent research suggests that line-labelling discrepancies may not be strongly related to medical errors or harm.[13] The project also encountered challenges relating to consensus on the chosen solution. The line-labelling policy was framed as a common-sense technical change, but this did not account for those areas that would need to restructure their work to accommodate the new standard,[1 24] particularly those who felt that the requirement to label all lines was in conflict with their perception that some lines did not need to be labelled. Providing a coherent programme theory for the intervention[25 26] that identified the goals and the mechanisms of change[25 27] might have helped staff to better understand the rationale behind the policy. Spending time on problem exploration and definition leads to better solution design,[28 29] and this is important when creating a case for change to convince others that there is a problem that needs to be worked on.

Further challenges arose relating to design problems that surfaced only after implementation. Important strategies for optimising and implementing design-based solutions include prototyping (circulating physical exemplars, ie, the new labels) and piloting (testing the proposed policy in a defined perimeter). These activities support communication with stakeholders, aid in learning and inform decision-making about prospective designs.[30] Prototyping, testing and evaluating solutions in short refinement cycles are core aspects of a design thinking approach.[31 32] When confronting potential users with prospective designs, they should be presented with an artefact with which they can interact, and observe how they do it—for example, through clinical simulation.[33 34] Observation is important because participants may estimate one solution to be better in a survey, when actual experimentation shows otherwise.[35] However, prototyping and piloting was very limited in the line-labelling project. One problem was that the new colour code for labels in some cases conflicted with established standards that appeared to be effective, pointing to the more general problem that standardisation that involves de-implementation of existing practices is often a fraught and delicate challenge.[36]

Finally, mundane supply chain issues such as capacity at the printing company and availability of budgets for printing plates compounded the challenges, affecting simultaneous implementation of the policy across the region. Anticipating needs and ensuring sufficient availability of labels before setting an implementation date

would have helped to prevent these delays. Models, such as process models or project management models, can be helpful to plan the various phases of projects and to design implementation processes.[37 38]

In terms of informing future practices, these findings illustrate general learning relevant for efforts to standardise in healthcare. In particular, emphasising that new standards are not implemented in a vacuum; they always have to fit into an ecology of pre-existing systems, norms, behaviours and established practices.[10] Standardisation is often considered as a neutral exercise, yet it may privilege the views of particular groups and alter how people work together in a sociomaterial world.[24] Understanding the underlying nature of these pre-existing elements before designing, let alone implementing, standards is crucial to ensure that standardisation is not seen as a merely technocratic process, but rather as one that takes into consideration the potential impact on working routines and relationships and recognises the political nature of standardisation.[1 24] This ecology affects the whole life cycle of any standard, from its inception to its deployment. All stages involve a network of stakeholders, specific constraints and distinct challenges. Delivering on operational reality for standardisation projects requires creating shared vision and understanding, strong coordination and project management to secure progress, and a 'design thinking' spirit with attention to multiple mundane details to deliver practical, usable standards. Relevant and ongoing communications with key stakeholders should be seen as good practice when developing standards. This also includes the need for appropriate testing or simulation of standards before they are implemented on a larger scale, particularly when changing or introducing the use of technologies or other physical artefacts in healthcare settings. This allows for adequate reflection on the proposed standards, including both physical designs and also the rationale for standardisation.

This study provided a useful case study from which to draw lessons for future large-scale standardisation projects. We were able to interview a broad range of stakeholders, though not as many people as originally planned, and the geographical spread of those interviewed was uneven. Further, the study was retrospective, based on interviews and documents, aiming to explore staff experiences of the design and implementation of the policy. Consequently, it was not possible to observe meetings or to collect data on the impact of the line-labelling project on compliance or outcomes.

## CONCLUSIONS

In this case study of the development and implementation of a standardised line-labelling policy in one of the devolved regional healthcare systems in the UK, what appeared to be a viewed as a 'common-sense' response to a perceived problem did not fully take into account the wider system, including established norms, practices and existing standards. Consequently,

not all staff were convinced of the need for this policy or of its proposed efficacy. These findings suggest that improving understanding of the underlying theoretical and philosophical aspects of standardisation, coupled with the introduction and use of 'design thinking' techniques, would improve future large-scale standardisation processes in healthcare.

**Acknowledgements** The authors thank the participants in the study. We thank Katrina Brown for her extensive comments on the paper prior to submission.

**Contributors** Obtained funding: MD-W. Study design: MD-W. Data analysis: NMK. Data interpretation and drafting of the manuscript: NMK and GL. Critical revision of manuscript: MD-W. All authors approved the final version of the manuscript.

**Funding** This work was supported by MDW's Wellcome Trust Investigator grant award WT09789. GL and MD-W are supported by Health Foundation's grant to the University of Cambridge for The Healthcare Improvement Studies (THIS) Institute. THIS Institute is supported by the Health Foundation – an independent charity committed to bringing about better health and health care for people in the UK. MDW is a National Institute for Health Research (NIHR) Senior Investigator (NF-SI-0617-10026).

**Disclaimer** The views expressed in this article are those of the authors and not necessarily those of the NHS, the NIHR, or the Department of Health and Social Care.

**Competing interests** None declared.

**Patient and public involvement statement** This was a study of a policy targeting staff behaviours and organisational systems. Patients and the public were not involved in this study.

**Patient consent for publication** Not required.

**Ethics approval** This study received ethical approval from the Cambridge Psychology Research Ethics Committee (Application No: PRE.2016.072).

**Provenance and peer review** Not commissioned; externally peer reviewed.

**Data availability statement** No data are available. The consent for this project restricts use of the transcripts to the study team. Requests to access anonymised data should be made to the corresponding author; further ethical and governance approvals may be required to enable sharing.

**ORCID iDs**
Natasha Marie Kriznik http://orcid.org/0000-0002-4291-0614
Guillaume Lamé http://orcid.org/0000-0001-9514-1890

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
