## [Reviewer comments · BMJ Open]

ARTICLE DETAILS

TITLE (PROVISIONAL)	Challenges in making standardisation work in healthcare: lessons from a qualitative interview study of a line-labelling policy in a UK region
AUTHORS	Kriznik, Natasha; Lamé, Guillaume; Dixon-Woods, Mary

VERSION 1 - REVIEW

REVIEWER	Dominic Furniss University College London, UK
REVIEW RETURNED	04-Jun-2019

GENERAL COMMENTS	I was interested to review this paper as we have done work in this area. I liked the paper but had a few issues I wanted to raise with the authors that might hopefully improve the paper: 1) I don't have a clear idea of what 'line labelling' is. I think this should be defined and all the example of different types of line labelling listed. For example, does it include enteral feeding, epidurals, intravenous infusions, etc. Are there examples of things that people might consider line labelling that are actually excluded? This is important for understanding the paper. There might be the further point that the standardised policy did not have a clear view of this definition and scope, e.g. it seemed nurses did not think dialysis should be included.2) The other term in the abstract and later in the paper is 'health system' which again I think needs further explanation. Is this a ward, a clinical area, a hospital or hospitals, is it something else?3) The related work that I referred to earlier has gathered data on intravenous infusion practice and errors. We found lots of 'tube tagging' discrepancies but this was not really related to errors and harm in our data (Lyons et al., 2018) – this relates to your point about acknowledging the fundamental problem you're trying to solve. Of course this does not mean that tube tagging could be related to significant medical error and harm. We explored some of the variability between policy and practice and 'tube tagging' was a frequent issue at some trusts in the sense that it was their largest proportion of discrepancies. However, many policies seem to differ in the details they required and how lines should be labelled. The most sensible seemed to say that IV lines should be labelled for continuous infusions (for infection control policies) and for multiple IV (to help differentiate them). This seemed more practical compared with requirements to label IV lines that would only be used for 20min. See Furniss et al. (2018). You can judge how best to integrate this work into your paper if it seems useful but it seems to support your arguments, e.g. the requirements to label
---

	some lines was in conflict with nurses perception of the value of doing so. Other minor things  1) 'or solve' in the abstract seems like an orphaned phrase 2) Highlights: Could we have more information on what 'regional' means? 3) Bottom of Intro section: Are we allowed to know the devolved nation? 4) Results: Only 3 trusts involved but out of how many, e.g. 4 or 40? 5) Pge 12 Ln10-12: I don't know if you want to make the point later that arguably studying these sources of variability and inconsistencies could be a good source of data for understanding where staff are having issues, where they see and do not see risks. 6) Pge 13 Ln14-20: Maybe consider adding a cross-reference to the doctor example you just spoke about, where the doctor knew it was a dialysis line and went ahead and used it anyway, so a label would not have helped. 7) An open question, and maybe something just to reflect on rather than put it the paper, but could resistance to this poorly designed policy be seen as a success, so money is not wasted in buying unnecessary labels and staff time and moral is saved in not doing things they see as unnecessary? I was inspired by these thoughts when you were alluding to outcomes on Pge 15. References Lyons, I., Furniss, D., Blandford, A., Chumbley, G., Iacovides, I., Wei, L., ... & Schnock, K. O. (2018). Errors and discrepancies in the administration of intravenous infusions: a mixed methods multihospital observational study. BMJ Qual Saf, 27(11), 892-901. Furniss, D., Lyons, I., Franklin, B. D., Mayer, A., Chumbley, G., Wei, L., ... & Blandford, A. (2018). Procedural and documentation variations in intravenous infusion administration: a mixed methods study of policy and practice across 16 hospital trusts in England. BMC health services research, 18(1), 270.
--	---

REVIEWER	Mark Jeffries University of Manchester, UK
REVIEW RETURNED	13-Aug-2019

GENERAL COMMENTS	Thank you for the opportunity to review this paper. Identifying and learning from the challenges of implementation of such standardised policies is an important area for patient safety and the manuscript was an interesting read. Overall I think this was well written and certainly has much potential. My main issues are with a lack of clarity – I make some specific suggestions below as to how this might be improved. Also what is demonstrated in the results is really interesting but I feel there is a bit of a missed opportunity in exploring the findings in relation to the “complex sociotechnical processes” mentioned in the introduction and the “ecology of pre-existing systems, norms, behaviours and established practices” mentioned in the conclusion. Some of this is explored in the discussion but I just feel that there could have been more explicit linking to these ideas. I think some restructuring in the discussion might help with that. Some specific points are below. I hope these are helpful. Abstract
---

1. This provides a clear overview of the paper. In the section on Strengths and Limitations the second statement seems more of an observation of what was done rather than a strength. If you mean that this mix of staff was a strength perhaps this needs to be stated more explicitly.

Introduction

2. Could you please expand on the sentence at Page 3, line 20 about how standardisation neglects “underlying, theoretical, sociocultural and philosophical aspects”. I think a little bit more here is important given your findings and discussion later.

3. Please expand upon the final sentence of the first paragraph. What do you mean by complex sociotechnical processes? Why is this important in the implementation of standards? Again I think it is important to develop this somewhat given your findings and discussion later.

4. Could you please provide more a little more detail about the safety issues associated with the labelling of catheters? You mention three types of potential error but are vague about “other complications”. I think your paper will be of interest to a broad range of academics some of whom will not have a clinical background.

Methods

5. Can you explain more clearly here the long duration of the research from April 2011 to Jan 2017? It is not clear if this was the time span of the implementation or the data collection?

6. At line 26 page 4 you state that contacts in the trusts were asked to identify staff. How were these contacts themselves identified? How were they known to the research team?

7. Could you please provide further detail of how consent to participate was obtained eg. was this written consent?

8. There are typos at line 35 – I assume the word “to” is missed out.

9. Were you given any reasons for non- participation? Please state these if they were known.

10. For clarity please add if the interviews were audio or video recorded.

11. The third paragraph relating to the documentary sources needs much more clarity and further explanation. From where, how and by whom, were these policy documents obtained. Which meeting minutes and presentations? This is important because of the first section of the results.

12. I assume the word “Analysis” has been omitted from the first sentence of paragraph 4 line 52.

13. Can you be more specific about the analysis please? Who carried this out? When was analysis undertaken? Was analysis concurrent with data collection? How were the documentary sources analysed? Please could you explain how you used Nvivo – I’m not sure if facilitated is the right word here. Nvivo can help organise and manage data but I’m not sure how it can facilitate. Sorry if this sounds pedantic.

Results

14. The second part of the results under “Challenges Encountered” I find interesting and is presented well with appropriate quotations and an interesting outlining of the themes. The first part around developing and implementing the policy is I think somewhat problematic. I don’t think it is particularly clear where the information has come from or how these sources were obtained (see my point 10 above). Could you please provide

	sources for this narrative description? There are sections here where the source of the claim made is unclear – for instance the last sentence of the first paragraph (lines 44-51 page 6.) I think it might also help the reader if there was a table of the documents that were analysed which include dates they were looked at, key findings etc. It might be worth considering placing this material about the development of the policy in a figure or as a separate box. I only offer that as a possible suggestion as a way of providing clarity here. 15. Page 11 line 57 to page 12 line 12 – These appear to be discussion points rather than a reporting of the results. I think the implications of what has been found here could be left for the discussion. Discussion 16. Overall, I think the discussion summarises the findings and makes connections to existing literature through a range of citations. However, I feel the connections to previous research need to be more explicit. Some rewriting here is necessary to not just summarise what has been found but place it clearly into context of both previous research and policy. 17. Could you please expand upon your strengths and weaknesses. 18. Page 17 line 27 Personally I don't think it the aim of qualitative research to provide generalisable findings so I don't see your single case study as a limitation as such. Perhaps you could instead discuss here how the findings of the case study might be transferable to other contexts and settings. 19. Page 17 line 34. Could you please expand upon why you did not undertake observations? 20. Please could you provide detail on what you consider the implications of your findings to be for policy and practice? Conclusions 21. I think your concluding paragraph about the implementation lessons learnt is important but feel this needs to be part of the discussion. 22. Please can you comment further on the “ecology of pre-existing systems, norms, behaviours and established practices” mentioned at line 12 page 18 particularly in light of your comments in the introduction and tie this more closely to your findings.
--	---

VERSION 1 – AUTHOR RESPONSE

Reviewer 1 Comments	Response / changes made	Location of change
Main comments		
1. I don't have a clear idea of what 'line labelling' is. I think this should be defined and all the example of different types of line labelling listed. For example, does it include enteral feeding, epidurals, intravenous infusions, etc. Are there examples of things that people might consider line labelling that are actually excluded? This is important for understanding the paper. There might be the further point that the standardised policy did not have a clear view of this	Wording added to expand on what line labelling means, and the kinds of lines that are included in the regional policy. We have now noted that there were issues of definition and scope that further complicated this programme.	Introduction, all paras

definition and scope, e.g. it seemed nurses did not think dialysis should be included.		
2. The other term in the abstract and later in the paper is 'health system' which again I think needs further explanation. Is this a ward, a clinical area, a hospital or hospitals, is it something else?	We have clarified that we are referring to a "devolved regional health system"	Abstract
3. The related work that I referred to earlier has gathered data on intravenous infusion practice and errors. We found lots of 'tube tagging' discrepancies but this was not really related to errors and harm in our data (Lyons et al., 2018) – this relates to your point about acknowledging the fundamental problem you're trying to solve. Of course this does not mean that tube tagging could be related to significant medical error and harm. We explored some of the variability between policy and practice and 'tube tagging' was a frequent issue at some trusts in the sense that it was their largest proportion of discrepancies. However, many policies seem to differ in the details they required and how lines should be labelled. The most sensible seemed to say that IV lines should be labelled for continuous infusions (for infection control policies) and for multiple IV (to help differentiate them). This seemed more practical compared with requirements to label IV lines that would only be used for 20min. See Furniss et al. (2018). You can judge how best to integrate this work into your paper if it seems useful but it seems to support your arguments, e.g. the requirements to label some lines was in conflict with nurses perception of the value of doing so. Lyons, I., Furniss, D., Blandford, A., Chumbley, G., Iacovides, I., Wei, L., ... & Schnock, K. O. (2018). Errors and discrepancies in the administration of intravenous infusions: a mixed methods multihospital observational study. BMJ Qual Saf, 27(11), 892-901. Furniss, D., Lyons, I., Franklin, B. D., Mayer, A., Chumbley, G., Wei, L., ... & Blandford, A. (2018). Procedural and documentation variations in intravenous infusion administration: a mixed methods	These are very useful comments. We have now referenced these very helpful articles.	Introduction, para 1 and 2; Discussion para 2

study of policy and practice across 16 hospital trusts in England. BMC health services research, 18(1), 270.		
Minor comments		
1. 'or solve' in the abstract seems like an orphaned phrase	'or solve' has been deleted.	Abstract
2. Highlights: Could we have more information on what 'regional' means?	Mentions of 'regional' have been amended to clarify that this is about a devolved region in the UK, and that the policy applies across healthcare organisations within this region only.	
3. Bottom of Intro section: Are we allowed to know the devolved nation?	We do not wish to identify the devolved nation.	
4. Results: Only 3 trusts involved but out of how many, e.g. 4 or 40?	We do not wish to provide exact numbers as that would reveal the devolved nation.	
5. Pge 12 Ln10-12: I don't know if you want to make the point later that arguably studying these sources of variability and inconsistencies could be a good source of data for understanding where staff are having issues, where they see and do not see risks.	The first part of that paragraph has been moved to the discussion (as suggested by reviewer 2), which seems to cover this point.	
6. Pge 13 Ln14-20: Maybe consider adding a cross-reference to the doctor example you just spoke about, where the doctor knew it was a dialysis line and went ahead and used it anyway, so a label would not have helped.	Added a cross reference to this point.	Results, Challenges encounter
7. An open question, and maybe something just to reflect on rather than put it the paper, but could resistance to this poorly designed policy be seen as a success, so money is not wasted in buying unnecessary labels and staff time and moral is saved in not doing things they see as unnecessary? I was inspired by these thoughts when you were alluding to outcomes on Pge 15.	Thank you for this suggestion. Although it is an interesting point, we feel it stretches the data somewhat so have not included it in the Discussion.	

Reviewer 2 Comments	Response / changes made	Location of change
Abstract		
1. This provides a clear overview of the paper. In the section on Strengths and Limitations the second statement seems more of an observation of what was done rather than a strength. If you mean that this mix of staff was a strength perhaps this needs to be stated more explicitly.	The second statement has been amended to clarify that our diverse sample was a strength: "We were able to recruit a mixture of policy-makers and frontline staff, which allowed us to understand a range of different experiences of the design and	Strengths and Weaknesses

	implementation of the regional line-labelling policy.”	
Introduction		
2. Could you please expand on the sentence at Page 3, line 20 about how standardisation neglects “underlying, theoretical, sociocultural and philosophical aspects”. I think a little bit more here is important given your findings and discussion later.	We have now better framed the Introduction and Discussion.	Introduction, Discussion
3. Please expand upon the final sentence of the first paragraph. What do you mean by complex sociotechnical processes? Why is this important in the implementation of standards? Again I think it is important to develop this somewhat given your findings and discussion later.	We have added further text to highlight the importance of considering existing systems and the interactions within them as a part of developing standards.	Introduction, paras 1 and 2
4. Could you please provide more a little more detail about the safety issues associated with the labelling of catheters? You mention three types of potential error but are vague about “other complications”. I think your paper will be of interest to a broad range of academics some of whom will not have a clinical background.	This section has been amended to highlight the key issue of wrong route administration, and possible causes for this.	Introduction, para 2
Methods		
5. Can you explain more clearly here the long duration of the research from April 2011 to Jan 2017? It is not clear if this was the time span of the implementation or the data collection?	“between April 2011 and January 2017” has been deleted to reduce confusion about the study duration. The study itself was conducted in 2017, while the historical process for the development of the regional line-labelling policy began in 2011 according to the documentary sources provided to the study team.	Methods
6. At line 26 page 4 you state that contacts in the trusts were asked to identify staff. How were these contacts themselves identified? How were they known to the research team?	The process for staff identification and contact has been amended to provide a clearer explanation.	Methods, para 2
7. Could you please provide further detail of how consent to participate was obtained eg. was this written consent?	Written consent was provided before interviews were conducted.	Methods, para 2
8. There are typos at line 35 – I assume the word “to” is missed out.	Thank you, we have corrected.	Methods, para 2
9. Were you given any reasons for non-participation? Please state these if they were known.	We did not have access to individuals who chose not to participate, so could	

	not establish reasons for non-participation.	
10. For clarity please add if the interviews were audio or video recorded.	Added in "audio" to final sentence of para 2 to clarify the interviews were audio recorded	Methods, para 2
11. The third paragraph relating to the documentary sources needs much more clarity and further explanation. From where, how and by whom, were these policy documents obtained. Which meeting minutes and presentations? This is important because of the first section of the results.	Additional details have been added to clarify the nature of these sources.	Methods, para 3
12. I assume the word "Analysis" has been omitted from the first sentence of paragraph 4 line 52.	Thank you, we have corrected this.	Methods, para 4
13. Can you be more specific about the analysis please? Who carried this out? When was analysis undertaken? Was analysis concurrent with data collection? How were the documentary sources analysed? Please could you explain how you used Nvivo – I'm not sure if facilitated is the right word here. Nvivo can help organise and manage data but I'm not sure how it can facilitate. Sorry if this sounds pedantic.	Added in clarification that the analysis was concurrent to data collection and who carried out the analysis. NVivo was used to organise and manage the data and coding.	Methods, para 4
Results		
14. The second part of the results under "Challenges Encountered" I find interesting and is presented well with appropriate quotations and an interesting outlining of the themes. The first part around developing and implementing the policy is I think somewhat problematic. I don't think it is particularly clear where the information has come from or how these sources were obtained (see my point 10 above). Could you please provide sources for this narrative description? There are sections here where the source of the claim made is unclear – for instance the last sentence of the first paragraph (lines 44-51 page 6.) I think it might also help the reader if there was a table of the documents that were analysed which include dates they were looked at, key findings etc. It might be worth considering placing this material about the development of the policy in a figure or as a separate box. I	Added in several references to documentary sources to support the narrative in the first section of the results. We have tried presenting this outline in a table previously, but found that having a narrative was much more effective in providing background and history to the development and implementation of the policy. There is the further challenge that providing precise details means that our case study can be more easily identified.	Results, section 1

only offer that as a possible suggestion as a way of providing clarity here.		
15. Page 11 line 57 to page 12 line 12 – These appear to be discussion points rather than a reporting of the results. I think the implications of what has been found here could be left for the discussion.	Thank you for this suggestion. This has been moved to the relevant point in the Discussion	
Discussion		
16. Overall, I think the discussion summarises the findings and makes connections to existing literature through a range of citations. However, I feel the connections to previous research need to be more explicit. Some rewriting here is necessary to not just summarise what has been found but place it clearly into context of both previous research and policy.	The Discussion has been reworked in order to make these links more explicit.	Throughout Discussion
17. Could you please expand upon your strengths and weaknesses.	We have expanded the discussion of the strengths and weaknesses.	Discussion, para 6
18. Page 17 line 27 Personally I don't think it the aim of qualitative research to provide generalisable findings so I don't see your single case study as a limitation as such. Perhaps you could instead discuss here how the findings of the case study might be transferable to other contexts and settings.	We now reflect on the fact that this study provides important insights into challenges of standardisation.	Discussion, throughout
19. Page 17 line 34. Could you please expand upon why you did not undertake observations?	We have now provided this explanation.	Discussion, para 6
20. Please could you provide detail on what you consider the implications of your findings to be for policy and practice?	The paragraph in the Conclusion has been moved into the Discussion and used as the basis for expanding on implications for future practices and approaches to standardisation.	Discussion, para 5
Conclusions		
21. I think your concluding paragraph about the implementation lessons learnt is important but feel this needs to be part of the discussion.	This has been moved into the Discussion as suggested.	
22. Please can you comment further on the “ecology of pre-existing systems, norms, behaviours and established practices” mentioned at line 12 page 18 particularly in light of your comments in the introduction and tie this more closely to your findings.	Additional text has been included to explain the importance of considering the context into which standards might be introduced.	Discussion, para 5

VERSION 2 – REVIEW

REVIEWER	Dominic Furniss Human Reliability Associates, UK.
REVIEW RETURNED	25-Oct-2019

GENERAL COMMENTS	I am satisfied with how the authors have dealt with the points raised in the previous round of review, and do not have further issues to raise.
---

REVIEWER	Mark Jeffries University of Manchester, UK
REVIEW RETURNED	29-Oct-2019

GENERAL COMMENTS	Thank you for your very detailed responses to my previous comments particularly with regard to the first part of the results and to the discussion which. The manuscript reads really well now and will be of great interest.
---